# Identifying the “Dangshan” Physiological Disease of Pear Woolliness Response via Feature-Level Fusion of Near-Infrared Spectroscopy and Visual RGB Image

**DOI:** 10.3390/foods12061178

**Published:** 2023-03-10

**Authors:** Yuanfeng Chen, Li Liu, Yuan Rao, Xiaodan Zhang, Wu Zhang, Xiu Jin

**Affiliations:** 1College of Information and Computer Science, Anhui Agricultural University, Hefei 230001, China; 2College of Horticulture, Anhui Agricultural University, Hefei 230001, China

**Keywords:** “Dangshan” pear, near-infrared reflectance spectroscopy, computer vision technology, feature fusion, disease identification

## Abstract

The “Dangshan” pear woolliness response is a physiological disease that causes large losses for fruit farmers and nutrient inadequacies.The cause of this disease is predominantly a shortage of boron and calcium in the pear and water loss from the pear. This paper used the fusion of near-infrared Spectroscopy (NIRS) and Computer Vision Technology (CVS) to detect the woolliness response disease of “Dangshan” pears. This paper employs the merging of NIRS features and image features for the detection of “Dangshan” pear woolliness response disease. Near-infrared Spectroscopy (NIRS) reflects information on organic matter containing hydrogen groups and other components in various biochemical structures in the sample under test, and Computer Vision Technology (CVS) captures image information on the disease. This study compares the results of different fusion models. Compared with other strategies, the fusion model combining spectral features and image features had better performance. These fusion models have better model effects than single-feature models, and the effects of these models may vary according to different image depth features selected for fusion modeling. Therefore, the model results of fusion modeling using different image depth features are further compared. The results show that the deeper the depth model in this study, the better the fusion modeling effect of the extracted image features and spectral features. The combination of the MLP classification model and the Xception convolutional neural classification network fused with the NIR spectral features and image features extracted, respectively, was the best combination, with accuracy (0.972), precision (0.974), recall (0.972), and F1 (0.972) of this model being the highest compared to the other models. This article illustrates that the accuracy of the “Dangshan” pear woolliness response disease may be considerably enhanced using the fusion of near-infrared spectra and image-based neural network features. It also provides a theoretical basis for the nondestructive detection of several techniques of spectra and pictures.

## 1. Introduction

Pear (*Pyrus* spp.) is a widely grown and consumed fruit [1,2]. There are many distinct types of pears, and they have varied properties [3,4]. Among them, the “Dangshan” pear is one of the most popular types in China [5]. The “Dangshan” pear is popular because it has the advantage of having thin and juicy skin. However, fruit chaffing disease of the “Dangshan” pear has caused large losses to fruit farmers, and this disease often occurs in “Dangshan” pear farming. Woolliness response disease is a physiological disease [6]. This disease is connected to a lack of nutrients or a reduction in root uptake in “Dangshan” pears, in which the deficient nutrients are largely boron, calcium, and water. In the absence of calcium, iron, and boron or in the presence of reduced root uptake, the fruit is encouraged to ripen quicker, and fruit hardness is reduced. This results in the development of woolliness response disease in the fruit.

To prevent the occurrence of woolliness response disease in “Dangshan” pears, effective and accurate detection methods have been researched, and strategies have been explored mainly around the causes of woolliness response disease. The traditional detection of mineral nutrients is largely based on laboratory physicochemical analysis, including inductively coupled plasma–mass spectrometry (ICP–MS), atomic absorption spectrometry, and UV-VIS spectrophotometry measures [7,8,9]. Although the results of these approaches are quite accurate, they have the disadvantages of destructive sampling and being time-consuming, labor-intensive, and costly, and these characteristics bring numerous restrictions to the study of mineral nutrition in pear fruit. Among the numerous non-destructive testing techniques, near-infrared reflectance spectroscopy (NIRS) and computer vision systems (CVS) are commonly utilized.

Several studies have employed NIRS and CVS techniques to diagnose diseases in many agricultural products [10]. For example, YanYu et al. [11] established a tool for monitoring fruit quality from a combination of NIRS and chemical models and then utilized this tool to explore the construction of a generic model that, among other things, predicts the Soluble Solids Content (SSC) of thin-skinned fruits with similar physicochemical properties. Lei-Ming Yuan et al. [12] employed vis-NIRS technology paired with a bias fusion modeling strategy for the noninvasive assessment of “Yunhe” pears, which improved the loss of spectral information in the optimized PLS model. Cavaco et al. [13] proposed a segmented partial least squares (PLS) prediction model for the hardness of “Rocha” pears in combination with a vis-NIRS segmentation model, which was used to predict the hardness of the fruit during ripening under shelf-life conditions. Pereira et al. [14] devised a color-image-processing method that employed a combination of digital photography and random forest to anticipate the ripeness of papaya fruits, evaluating them on the basis of flesh firmness. Zhu et al. combined process characteristics and image information to evaluate the quality of tea leaves [15]. Shumian Chen et al. developed a machine vision system for the detection of defective rice grains [16].

Near-infrared spectroscopy (NIRS) detection methods have the advantage of being nondestructive, convenient, environmentally friendly, and safe [17]. The method’s coverage of molecular absorption covers frequency combinations and double-frequency absorption of hydrogen-containing groups or other chemical bonds in many organic compounds, mainly C-H, N-H, S-H, O-H, and others. Therefore, the spectral profile produced using NIR spectroscopy may then be utilized to better reflect information on the organic matter of the hydrogen-containing groups in the sample under test and information on the composition of several other biochemical structures [18]. Machine vision technology has the advantages of being nondestructive, rapid, efficient, and objective [19,20]. This technique uses optical systems and image-processing equipment to replicate human vision. The technology extracts information from the acquired target image and processes it to obtain the information required for the object to be detected and then analyzes it. Therefore, the image information produced by machine vision technology can properly and objectively depict the appearance features of “Dangshan” pears, and this advantage plays a significant part in the woolliness response disease of “Dangshan” pears.

These studies detected diseases based on only a single aspect of NIRS or CVS. However, the NIRS and CVS approaches can only acquire the main components and appearance of the samples separately and cannot obtain complete information on the quality of “Dangshan” pears. Therefore, if NIRS features or CVS features are fused for disease diagnosis, this may improve the accuracy of diagnosing woolliness response disease. Feature-level fusion procedures make it possible to study sample features fully, and several studies have validated the use of feature-level fusion [21,22]. At the same time, this research used several forms of data-fusion algorithms to merge information from multiple detection techniques to produce better sample characterization and enhanced identification. Therefore, to effectively diagnose the woolliness response disease of “Dangshan” pears, researchers have concentrated on multi-technology integration to address the drawbacks of utilizing a single form of technology. Studies have performed data fusion by merging data from numerous different sources. A hybrid method was devised by Miao et al. to distinguish nine species of ginseng [23]. Fun-Li Xu et al. [24] used CVS and HSI techniques to accomplish quick, nondestructive detection of frostbite in frozen salmon fillets.

Currently, in integrated spectroscopic and image techniques, spectral detection methods are generally used in hyperspectral techniques. Hyperspectral imaging (HSI) provides images in which each pixel contains spectral information to reflect the chemical properties of a particular region [25]. However, we discovered a few studies integrating NIRS and CVS for disease detection in fruit. Hyperspectral image techniques have disadvantages such as high cost and limited accessibility, which have become a not insignificant impediment in the development of disease diagnostic approaches for “Dangshan” pears. NIRS technology has the advantages of low cost and ease of field use [26,27]. Therefore, this work combines the application of NIRS and CVS systems with a feature-level fusion strategy to explore a method that can increase the accuracy of illness identification in “Dangshan” pears.

This study created a nondestructive, objective, and accurate approach for the diagnosis of chaffing diseases in “Dangshan” pears. The method incorporates a combination of an NIRS system and a CVS. We extracted the NIRS characteristics by modeling the machine-learning approach. CVS features are retrieved by employing a convolutional neural network. We then compare the classification results obtained using only NIRS features, only CVS features, and a fusion of NIRS and CVS features to determine the best method for “Dangshan” pear woolliness response disease identification and to explore and analyze the effects of different-layer fusion modeling for the CVS feature model. This study intends to provide a theoretical basis and innovative concepts for developing innovative technologies for the woolliness response disease of “Dangshan” pear.

The remainder of this study is separated into three sections. In Section 2, the spectral and image-data-gathering methods for “Dangshan” pears and the accompanying machine-learning methods, deep-learning methods, and feature-level fusion approaches are discussed. Section 3 covers not only the performance of the single-feature model and the feature-level fusion model but also evaluates and analyses the fusion effects of different layers of the convolutional neural network. Section 4 discusses the conclusions of this investigation.

## 2. Materials and Methods

### 2.1. Samples

In this investigation, 480 samples of “Dangshan” pear trees were collected and classified for this area in Yeji District, Liuan City, Anhui Province. The plants in this area have the advantages of uniform growth and robustness. However, the unpredictable weather in the region has led to frequent natural disasters such as floods and droughts in the area. As a result, local agricultural production has also suffered greatly.

In early September 2022, some of the “Dangshan” pear trees in the test site developed obvious chaffing symptoms and were called sick trees. The other section of the test location exhibited no evidence of chaffing, and the fruit was in normal condition; thus, these trees were called healthy trees. One diseased tree and one healthy tree of “Dangshan” pear were meticulously selected in the test site, 240 infected fruits were picked from the diseased tree, and subsequently, 240 normal fruits were picked from the healthy tree. As shown in Figure 1, the main symptoms of the disease are dark yellow fruit color, reduced fruit hardness, and symptoms spreading inwards from near the skin. After completing the picking, the fruit was carried back to the laboratory to be utilized as test samples. The surface of the pear fruit was washed and wiped clean before being numbered for use.

### 2.2. Data Acquisition Instruments

As shown in Figure 2a, the spectrum data were acquired using a handheld miniature NIR spectrometer in the spectral range of 900–1700 nm with 228 bands, a spectral resolution of 3.89 nm, and a signal-to-noise ratio (SNR) of 5000:1. The product model of this instrument was the NIR-S-G1, which was created by Shenzhen Puyan Network Technology Co., Ltd. (Shenzhen, China). Before collecting the spectral data, the instrument needed to be connected via Bluetooth to the app “Instagram” on a mobile phone. Before each “Dangshan” pear measurement, the spectrometer was calibrated with a standard white and dark reference. The instrument was placed close to the calibration whiteboard, and the light emitted shone on the whiteboard and was reflected into the spectrometer, which captured and recorded the brightness value (*W*) of the whiteboard; we turned off the emitted light of the instrument and recorded the brightness value (*B*) on the blackboard. After calibration, the instrument was utilized to gather spectral data on the surface of the pears, with the instrument’s light source window close to the “Dangshan” pear sample to obtain the reflected light being recorded as the luminance value (*R*) on the pear surface. The spectral reflectance of the sample was determined using Equation (1).
(1)$$R=\frac{\left(I-B\right)}{\left(W-B\right)}\times 100\%$$

As shown in Figure 2b, the equipment used to collect image data in this test was an “EOS 90D” digital camera manufactured by Canon Inc. (Tokyo, Japan). Canon is a leading Japanese integrated group that produces imaging and information products worldwide. The camera is an autofocus/auto-exposure single-lens reflex digital camera with a built-in flash. It contains a total of 34.4 million pixels and has a CMOS sensor type and a maximum resolution of 6000 × 4000.

The spectral-data-gathering method is presented in Figure 3a. Before the spectral data was collected, an ellipse was defined with a pencil at the equator on the surface of the typical fruit rind; the short axis of this ellipse was approximately 3 cm, and the long axis was approximately 5 cm. The area within this ellipse was utilized as the range for spectral-data acquisition; the ends of the central axis of the area and the center section were used as the range for spectral acquisition. The scanning window on the front of the compact handheld spectrometer was placed immediately within the specified range, followed by five scans in each sample area. At the end of the scan, each data file was named according to the sample number. Finally, the reflectance spectral data averaged over the five scans of each area were used as the original modeling spectral data.

The image-data-acquisition method is presented in Figure 3b. The procedure of image-data gathering was to position the “Dangshan” pear on a white background under natural light; the camera was kept at the same height as the “Dangshan” pear and approximately 30 cm distant from the front of the “Dangshan” pear. The image resolution was set to RAW (6960 × 4640), approximately 32.3 megapixels; the focus mode was single autofocus, the scene mode was macro, the sensitivity was still-image shooting, the parameters were established, and the shooting began. The normal fruit was photographed as a normal sample, and the infected fruit was photographed as a diseased sample.

### 2.3. Machine-Learning Methods for Near-Infrared Spectroscopy

Partial Least Squares Discriminant Analysis (PLSDA), support vector machines, random forests, and Boost-like approaches are utilized mainly in supervised machine-learning models for NIR spectroscopy. Among these, PLSDA technique is a chemometric tool that utilizes a PLS algorithm in modeling differences between defined sample classes, as such, allowing for the discrimination of samples within these groups [28]. SVM is a frequently used supervised classification-learning algorithm [29]. Its basic idea is to identify the most recognizable hyperplane by maximizing the edge distance between the nearest points in each class. RF (random forest) is an algorithm that combines Breiman’s bagging idea [30] with Ho’s random subspace approach [31]. They are generated based on decision trees that are trained using segments of the dataset and randomly selected segments of the feature set. AdaBoost (short for adaptive boosting) is based on the premise that a set of weak classifiers can yield a strong classifier. In this scenario, the weak classifiers are combined linearly but modified by the coefficients gained during training. The selection of weak classifiers focuses on examples that are more challenging to categorize. In this repeating procedure, the coefficients of the weak classifiers correspond to the classifier errors on the dataset. The coefficients of the weak classifiers with the fewest errors are enhanced. Strong classifiers group all these weak classifiers based on significance coefficients. XGBoost is a boosting method that transforms weak learners into strong learners [32]. As a boosted tree model, XGBoost is a powerful classifier consisting of many single tree models.

In addition, at the model level, one of the generally used modeling approaches is the multilayer perceptron (MLP), an artificial neural network model (ANN) consisting of many layers for which the network structure might be a feed-forward or feedback network structure. The model comprises an input layer, a hidden layer, and an output layer [33]. The MLP with one hidden layer is the simplest model with a network structure. The structure of an MLP involves the number of layers, the number of neurons, the transfer function of each neuron, and how the layers are coupled, depending on the type of problem [34]. Each neuron of an MLP has its own weight. A neuron can have any number of inputs from 1 to n, where *n* (an integer) is the total number of inputs. The inputs are denoted as $x_{1},x_{2},x_{3}\ldots x_{n}$; the corresponding weights are denoted as $w_{1},w_{2},w_{3}\ldots w_{n}$; and the output is represented as a = $x_{1}w_{1},x_{2}w_{2},x_{3}w_{3}\ldots x_{n}w_{n}$. The simple structure of node I in the MLP with k inputs from nodes {1,2,…*k*} with k input arcs from nodes {1,2,…k}, for which the associated weights and input values are $w_{1i}\ldots w_{ki},\;\mathrm{and}\;x_{1i}\ldots x_{ki}$. The dashed lines show the values propagated through the network. Predictions are shown by the $y_{i}$ values. Different activation functions ($f_{i}$) are applied to the input values to flow through the network. In the later stage of feature fusion, we designed the MLP with simply an input layer and a hidden layer for NIR spectral feature extraction, with weights initialized using a uniform distribution. The output layer utilizes the sigmoid activation function, and the remaining layers use the ReLU activation function.

### 2.4. Deep Neural Network Methods

CNNs have a vital role and significance in the field of computer vision. The structure of a convolutional neural network (CNN) has a standard structure consisting of alternating convolutional layers and a pooling layer following the convolutional layers. Then, based on the standard structure, the CNN performs fully connected classification [35]. In this scenario, the fully connected classification is performed by a fully connected output layer and a SoftMax classifier. The fully connected output layer is equivalent to a simple logistic regression using the equivalent of a standard MLP but without any hidden layers. SoftMax classifiers are typically trained using a backpropagation algorithm to find the weights and biases that minimize a certain loss function and then to map any input as close as possible to the target output. The two-dimensional convolution operation of a CNN is formulated as follows:(2)$$X_{k}^{l}=f\left(\sum_{i\in M_{j}}X^{i-1}*k_{ij}^{l}+b_{j}^{l}\right)$$
where $X_{k}^{l}$ is the kth feature map of layer l, $M_{j}$ represents the input map part, $k_{ij}^{l}$ is the learnable kernel, f(⋅) represents the activation function, and $b_{j}^{l}$ is the bias term of the kth feature map of layer l.

Since this study will examine and analyze fusion models for distinct layers, we will execute image-feature extraction for the given layers as input to the feature-fusion process.

CNNs are commonly used due to their excellent effectiveness. Some of the popular architectures studied in this study are discussed below. VGGNet is a deep architecture that can extract features at low spatial resolution [36].There are two variants of VGGNet: VGG16 (with 16 weight layers) and VGG19 (with 19 weight layers). The VGG16 architecture contains 16 weight layers. It also contains five convolution blocks and two fully connected layers. VGG19 differs from the VGG-16 model in that it is planned to start with five convolution blocks followed by three fully connected layers. Similar to VGG16, these convolutional layers utilize a 3-kernel with a step size of 1 and a padding of 1; therefore, the dimensionality of the feature mapping will be the same as that of the previous layers.

A deep residual network named ResNet was proposed by He et al. [37]. ResNet consists of sequentially ordered convolutional, pooling, activation, and fully connected layers. ResNet includes multiple architectures, including ResNet 50 and ResNet 101. ResNet 50 comprises a backbone for input, four stages, and an output layer [38].

The ResNet101 network differs somewhat from ResNet50 in that there are 104 convolutional layers in the ResNet101 network, and there are 33 block layers alongside them. In addition, the output of the previous block is used as the residual connections directly above these 29 blocks. To receive the input from the other blocks, these residual connections are exploited at the termination of each block using the starting value of the sigma operator. The ResNet101 network is organized into six basic sections, namely, the input module, four different structured blocks, and the output block. The building blocks of the network model are essentially residual block structures. Each layer of ResNet101 uses the ReLU activation function and incorporates batch normalization units to improve the adaptability of the model. The ADAM optimizer was also employed to improve the accuracy of the network recognition. In the later stage of fusion modeling of different layers, we divided ResNet101 into five layers (Layer 1, Layer 2, Layer 3, Layer 4, and Layer 5) for image-feature extraction of different layers. The parameters and 2D output dimensions of different layers of the ResNet101 network are provided in Table 1.

The Xception architecture, which was developed by Francois [39], is a linear stack of deeply separable convolutional layers with residual connections [40]. It is an upgraded model of InceptionV3.

One of the building blocks of the Inception network is the Inception module, which collects parallel routes with differing perceptual field widths and actions in a feature-mapping stack [41].

After the amazing success of the Inception network, GoogLeNet (Inception V1) was changed into InceptionV2, Inception V3, and Inception-ResNet. Xception consists of 36 convolutional layers separated into three primary flows, namely, Entry flow, Middle flow, and Exit flow. Images from the training set are first transmitted to the Entry flow, which builds feature maps. These feature maps are then further input into the Middle flow (repeated eight times). Finally, the feature maps of the Exit flow create 2048-dimensional vectors. Xception substitutes the typical initial module with a depth-separable convolution (separate spatial convolution on each input channel), followed by point-specific convolution (1 × 1 convolution) [39]. In a later stage of fusion modeling of distinct layers, we divided Xception into three layers (Entry flow, Middle flow, and Exit flow) for image feature extraction in different layers. The architecture of the many layers of Xception is represented in Figure 4.

DenseNet201 is a convolutional neural network with a depth of 201 layers [42]. Each of these layers is connected using a feed-forward technique. The feature maps of all previous layers are utilized as input to each layer, and their feature maps are used as input in all subsequent layers. This architecture has a dense connectivity structure, hence the name dense convolutional neural network. DenseNet201 increases feature propagation, stimulates feature reuse, and significantly reduces the number of parameters [43].

### 2.5. Near-Infrared Spectroscopy and Visual Image Feature Fusion Methods

In this work, a feature-level fusion strategy was investigated, such that we applied spectral and image feature depth fusion methods for spectral and shape characteristics to diagnose woolliness response disease in “Dangshan” pears. A description of the spectral and image feature depth fusion method is presented in Figure 5.

The extraction of spectral features is divided into two steps. (1) The procedure of spectral data acquisition is the average of reflectance spectral data from five scans of each region of the sample as the original modeled spectral data. (2) The input layer of the MLP takes the spectral data as input and performs the extraction of NIR spectral features as input features for feature-level fusion.

The image features are extracted in four steps. (1) The image acquisition process is as follows. The image is taken by placing the camera approximately 30 cm in front and keeping it on a horizontal plane with the fruit, and the image is used as the raw image data. (2) The input layer of the CNN is connected to the image sample as the input signal. (3) Local features are extracted via convolutional operations of local perceptual fields in the CNN convolutional layer. CNN extracts the image signal by reducing duplicate information, thereby revealing the potential information in the image signal. (4) The image features extracted from the different layers of the network are output, which are subsequently used as input features for feature-level fusion.

The details of the feature-level fusion are as follows: the image feature vector is extracted via the CNN feature-extraction method, and then the spectral feature vector is extracted via the NIRS feature-extraction method. The extracted spectral feature map is a one-dimensional vector. The extracted image feature map is placed in a flattening layer and turned into a one-dimensional vector. The equation for the flattening operation is as follows.
(3)$$\left(Height,Width,Channel\right)\to \left(Height\times Width\times Channel\right)$$

The one-dimensional vectors are then stitched together to obtain the fused data with the equation shown below:(4)$$F_{f}=Append\left(F_{a},F_{b}\right)$$
where $F_{f}$ denotes the merged data, $F_{a}$ denotes the 1D feature map of the image sample, $F_{b}$ denotes the 1D feature map of the spectrum, and $Append\left(\cdot \right)$ denotes the fusion function that stitches the $F_{b}$ vector after the $F_{a}$ vector.

In this work, the feasibility of near-infrared reflectance spectroscopy (NIRS) features and image features for diagnosing woolliness response diseases in “Dangshan” pears was verified. The convolution depth of the visual image and the number of neurons in the spectral neural network were both analyzed and explored. Finally, the feasibility of the feature-level fusion technique was verified for NIRS features and image features, and the influence of fusion models on different layers of the network was explored and analyzed for convolutional gods.

### 2.6. Evaluation

This research employs confusion matrices to compare the performance of various networks, which reveal the number of accurate and incorrect predictions for each class in a given dataset. In addition, the performance of the model is tested in this study using four metrics—accuracy, precision, recall, and F1 (the cumulative average of precision and sensitivity)—according to the following equations.
(5)$$Accuracy=\frac{TP+TN}{TP+TN+FN+FP}$$
(6)$$Precision=\frac{TP}{TP+FP}$$
(7)$$Recall=\frac{TP}{TP+FN}$$
(8)$$F1=\frac{2TP}{2TP+FP+FN}$$
where TP (true positive) is the number of positives correctly classified; TN (true negative) is the number of negatives correctly classified; FP (false positive) is the number of negatives incorrectly classified; and FN (false negative) is the number of positives incorrectly classified.

## 3. Results and Discussion

### 3.1. Division of the Training and Validation Sets

In this study, the initial data comprised the following two types: healthy and diseased. Table 2 displays the number of these samples, with a total of 480 sample data. The sample set was divided based on a 7:3 ratio, where 70% (336 sample data points) was used to train the network, and the remaining 30% (144 sample data points) was utilized to validate the trained network. This experimental device has 32 GB of memory, and the experiments utilize the TensorFlow framework to build deep-learning model structures and run on an NVIDIA RTX 3060 GPU platform in the python environment.

### 3.2. No-Fusion Separate Modeling Evaluation

The number of hidden layer nodes in MLP is a structurally sensitive parameter, which means that a very low number of nodes may lead to poor training, while a high number of nodes might lead to overfitting [32]. Therefore, we picked ten groups of hidden layer nodes from 10 to 100 in 10 steps of traversal to develop the MLP model matching the number of hidden layer nodes. The MLP model was denoted as MLP X, where X specifies the number of hidden layer nodes. The validation set traversal results of the classification models with varied numbers of hidden layer nodes are given in Figure 6. When the MLP classification models built with different numbers of hidden layer nodes are compared, the results show that model MLP_90, which was built with 90 hidden layer nodes, had the best fit.

To validate the feasibility of spectral features to diagnose woolliness response diseases in “Dangshan” pears, we employed a support vector machine (SVM), random forest (RF), extreme gradient boosting (XGBoost), adaptive boosting (AdaBoost), and other machine-learning methods to develop models. A grid search approach is utilized to select the best hyperparameters for these machine-learning models. Details of the final parameters of the machine-learning models are presented in Table 2.

The validation set results for all machine-learning models are provided in Table 3. By comparing the validation set results of these models, it is shown that MLP had the best modeling performance. The ideal MLP model MLP_90 has accuracy (0.611), precision (0.614), recall (0.611), and F1 (0.608) on the validation set. The results reveal that MLP significantly outperformed other machine-learning models. MLP_90 was the best among the classification models generated using MLP with varied numbers of hidden layer nodes.

In the extraction of image features, this paper uses VGG16, VGG19, ResNet50, ResNet101, Xception, and DenseNet201 to perform comparisons with other models. The optimum model for image feature classification models is explored and analyzed. Migratory learning methods such as freeze layers and fine-tuning are employed to extract significant features. In this study, fine-tuning of transfer learning was employed to enhance the efficacy of the CNN architecture and replace the last layer of the pretrained training-only model. The training accuracy curves for the six image feature classification models are given in Figure 7a, with DenseNet201 converging as the fastest and the slowest model being VGG19. The training loss curves in Figure 7b reveal that DenseNet201 had the least training loss. The validation accuracy curves are given in Figure 7c, with ResNet101 converging the fastest and VGG19 converging the slowest. The validation loss curve in Figure 7d demonstrates that ResNet50 had the lowest value of validation loss.

The results of the six image feature classification models on the validation set are reported in Table 4. These results suggest that the Xception model outperformed the other models on the validation set in terms of accuracy (0.840), precision (0.879), recall (0.840), and F1 (0.836). The results reveal that Xception works best among the six convolutional neural network methods in this paper.

In summary, this section validates the feasibility of spectral features and image features in recognizing woolliness response diseases of “Dangshan” pears. The results of the classification models produced by the two features were compared and analyzed. Among the classification models for spectral features, MLP was the best model with the highest accuracy, precision, recall, and F1. Among these, MLP_90 was the best model with the highest validation accuracy among the models developed with the varied number of hidden layer nodes of MLP.

The spectral data used in this study resulted in unsatisfactory extraction of spectral features due to light-scattering effects [44]. Although effective preprocessing can essentially eliminate the light-scattering effect, finding the best spectral preprocessing method for different models is a complex process. To solve this challenge of how to choose the best preprocessing method, in the algorithm proposed in this paper, the spectral data are selected without preprocessing, and the spectral features extracted from the original spectral data are used directly, and then the spectral features are fused with the image features for modeling, thus enabling an end-to-end approach. Therefore, all six models outperformed the spectral feature classification model in terms of validation in the classification model of image features. The training accuracy curves of the six models were more or less the same, with the training accuracy exceeding 90%. In addition, the validation accuracy of all six models was above 75%. According to a detailed examination of the results, Xception had the best validation accuracy while maintaining the best training accuracy and training loss. This implies that the Xception model is the best model among the six image feature classification models and has greater learning ability for the identification of woolliness response diseases in “Dangshan”.

### 3.3. Modeling of Spectral and Image Fusion Features

In this study, the image feature vector was first extracted using the CNN feature extraction method, and then the spectral feature vector was extracted using the NIRS feature extraction method. The two features were then concatenated to generate a multidimensional vector. This vector was used as the input to the prediction layer. The final output of the prediction layer was employed as the score of two pear classes, where the class with the greatest score was regarded as the acknowledged class of pears. The spectral feature model used MLP to extract NIR spectral features, and the image feature network model employed DenseNet201, ResNet50, ResNet101, VGG16, VGG19, and Xception migration learning models to extract image features.

This study compared and analyzed the model effects of combining different spectral and image models for fusion modeling. In this case, the spectral models were MLP models built with different numbers of hidden layer nodes, and the image models were different convolutional neural classification networks. In this paper, ten different models of MLP (MLP_10, MLP_20, MLP_30, MLP_40, MLP_50, MLP_60, MLP_70, MLP_80, MLP_90, and MLP_100) were selected to determine the most suitable MLP models for fusion.

The classical ResNet and VGG were then used in this study to initially identify the most suitable MLP models for fusion. The accuracy of the training model is not necessarily positively correlated with the number of model layers. This is because as the number of network layers increases, the network accuracy appears to saturate and decreases. Therefore, VGG16 is preferred between the two models VGG16 and VGG19; ResNet50 is preferred between the two models ResNet50 and ResNet101. In this subsection, VGG16 and ResNet50 were used to model the fusion with different MLP models, where the model with the best results was the most suitable model. The accuracy and F1 of the fusion modeling of the two convolutional network models with the MLP are shown in Figure 8. The results show that MLP_30_VGG16 and MLP_30_ResNet50 had the best validation results with the highest accuracy and F1. Therefore, the MLP_30 model with 30 nodes in the hidden layer is the most suitable model for feature fusion.

After the ideal number of hidden layer nodes was determined for the MLP model to be 30, the optimal model combining NIR reflectance spectral features and image feature fusion modeling was further examined. The training accuracy curves of the six fusion models are given in Figure 9a, with MLP_30_VGG19 converging the fastest and MLP_30_VGG16 the slowest. The value of the latter was the worst. The training loss curve in Figure 9b shows that MLP_30_VGG19 had the smallest training loss value. The validation accuracy curves are given in Figure 9c, with MLP_30_ResNet101 converging the fastest and MLP_30_DenseNet201 converging the slowest. The validation loss curve in Figure 9d reveals that MLP_30_Xception had the lowest value of validation loss.

The results of the six fusion models on the validation set are reported in Table 5. Among these, the best modeling results were discovered for MLP_30_Xception, which had the highest accuracy (0. 972), precision (0. 974), recall (0. 972), and F1 (0. 972). In addition, MLP_30_ResNet101 also performed well, with good accuracy (0. 965), precision (0. 966), recall (0. 965), and F1 (0. 965). The results reveal that the combination of MLP and ResNet101 is the superior combination, and the combination of MLP and Xception is the best combination.

### 3.4. Optimization of Fusion Models for Different Depth Feature Layers of Visual Images

MLP shows outstanding performance with ResNet101 and Xception for fusion modeling. However, utilizing image features extracted from different layers of the network and fusing them will cause the models that they build to have different modeling effects. Therefore, this study uses ResNet101 and Xception to further evaluate the performance of convolutional neural networks with different layers for fusion modeling. Five alternative sets of layers (layer 1, layer 2, layer 3, layer 4, and layer 5) of ResNet101 were selected for feature-level fusion with MLP_30, and five models were built to explore the best-fused layers of ResNet101. The distinct fusion layer models are named MLP_30_ResNet101_X, where X is the different layers of ResNet101. At the same time, three alternative sets of layers of Xception (Entry flow, Middle flow, and Exit flow) were selected to be fused with MLP_30 at the feature level, and three models were built to explore the best fusion layers of Xception. The different fusion layer models are named MLP_30_Xception_Y, where Y is the different layers of Xception.

The training accuracy curves of ResNet101 for five alternative sets of layer fusion modeling are given in Figure 10a, with all five models obtaining training accuracies above 85%. The validation accuracy curves in Figure 10b show that MLP_30_ResNet101_layer 5 had superior recognition results compared to the other four models. The training accuracy curves for the three major process fusion models of Xception are presented in Figure 10c, with all three models having training accuracies of over 95%. The validation accuracy curves in Figure 10d show that MLP_30_Xception_Exitflow had superior recognition results compared to the other two models.

The results of simulating the fusion of multiple layers of the convolutional network with spectral features are displayed in Table 6. By comparing the results of the feature-level fusion of MLP_30 with five different layers of ResNet101 and three different processes of Xception, correspondingly, it was discovered that the MLP_30_ResNet101_layer5 model had the highest accuracy (0.917), precision (0.920), recall (0.951), and F1 (0.917). Precision (0.920), recall (0.951), and F1 (0.917). Additionally, for comparison, the MLP_30_Xception_Exitflow model was determined to have the highest accuracy (0.951), precision (0.956), recall (0.951), and F1 (0.951). The results show that MLP_30_ResNet101_layer 5 had better recognition than the other four models of ResNet101, while MLP_30_Xception_Exitflow had better recognition than the other two models of Xception.

In summary, this section verifies that the models created with different layer functions in ResNet101 and Xception had varying performances. Among these, MLP_30_ResNet101_layer5 had the best model performance with the highest validation accuracy among the five sets of different layer fusion models of ResNet101. MLP_30_Xception_Exitflow had the best model performance with the highest validation accuracy among the three primary process fusion models of Xception.

### 3.5. Optimal Model Analysis and Comparison

For the comparison in this section, we selected the optimal models MLP_90 and Xception among the spectral and image feature classification models and selected the two superior models MLP_30_ResNet101_layer5 and MLP_30_Xception_Exitflow for the fused features of infrared spectral features and image features. The accuracy comparison of the four models on the validation set is provided in Table 7. Among these, the MLP_30_Xception_Exitflow model, which used a feature-level fusion technique, exhibited good performance with the greatest accuracy (0.951), precision (0.956), recall (0.951), and F1 (0.951).

The classification confusion matrix for the four optimal models is presented in Figure 11. The results showed that the MLP_30_Xception_Exitflow network model had the highest validation accuracy. In the MLP_90 network model, 22 “Dangshan” pear diseased samples were incorrectly predicted to be “Dangshan” pear healthy samples, and 34 “Dangshan” pear healthy samples were incorrectly predicted to be “Dangshan” pear diseased samples. In the Xception network model, 23 “Dangshan” pear healthy samples were incorrectly predicted to be “Dangshan” pear diseased samples. Three samples of “Dangshan” pear diseased in the MLP_30_ResNet101_layer5 network model were incorrectly predicted to be “Dangshan” pear healthy samples, and nine samples of “Dangshan” pear healthy samples were incorrectly predicted to be “Dangshan” pear diseased samples. In the MLP_30_Xception_Exitflow network model, seven samples were incorrectly predicted to be healthy samples.

In summary, in the comparison of the four superior models, MLP_90, ResNet101, MLP_30_ResNet101_layer5, and MLP_30_Xception_Exitflow, the MLP_30_Xception_Exitflow model, after employing the feature-level fusion technique, obtained the best classification results. The MLP_30_Xception_Exitflow model had the highest accuracy (0.951), precision (0.956), recall (0.951), and F1 (0.951). The combination of the MLP classification model and the Xception convolutional neural classification network with the fusion of the NIR spectral features and image features extracted separately was the best combination.

## 4. Conclusions

The fast and precise diagnosis of “Dangshan” pear woolliness response disease is vital, as it is a physiological disease that has a substantial impact on the quality of “Dangshan” pears. This research indicates that it is feasible to apply near-infrared reflectance spectroscopy (NIRS) features and image features to diagnose woolliness response disease in “Dangshan” pears. The experiments first acquired information on the chemical composition and appearance of the samples via NIRS and CVS techniques, respectively, and then used machine-learning and deep-learning methods for diagnostic classification. These findings imply that the feature-level fusion technique can utilize the advantages of NIRS and CVS features to gain more extensive sample information compared to single-feature models and consequently improve the accuracy of recognizing “Dangshan” pear diseases to a greater extent. Then, to explore the effect of different depth feature layers of visual pictures on fusion modeling, experiments were performed to model the fusion of CVS features extracted via different layers of convolutional neural networks with NIRS features. The results show that the fusion modeling of the feature layer with the highest depth of the visual image has a more accurate classification performance. In this study, the combination of the MLP classification model and the Xception convolutional neural classification network fused with the NIR spectral features and image features extracted, respectively, was the best combination, with the accuracy (0.972), precision (0.974), recall (0.972), and F1 (0.972) of this model being the highest compared to the other models. In summary, the use of near-infrared reflectance spectroscopy (NIRS) features and image feature fusion for the identification of “Dangshan” pear woolliness response disease is a promising method for disease diagnosis and provides a broad perspective in the field of fusion for agricultural disease diagnosis. It can provide new ideas for achieving fast, reliable, and nondestructive quality control instruments for various agricultural products.

## Figures and Tables

**Figure 1 foods-12-01178-f001:**
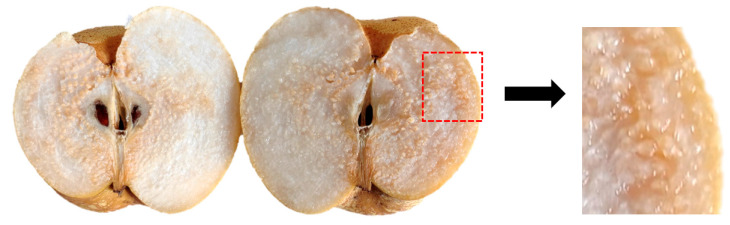
Diseased sample.

**Figure 2 foods-12-01178-f002:**
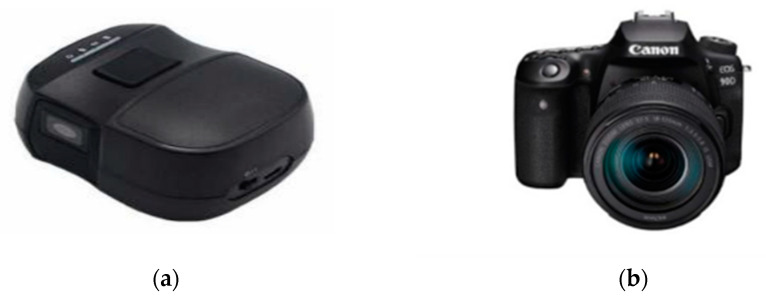
Instruments utilized for spectral data acquisition in the experiment: (**a**) “NIR-S-G1” miniature spectral acquisition apparatus; (**b**) “EOS 90D” single-lens reflex digital camera.

**Figure 3 foods-12-01178-f003:**
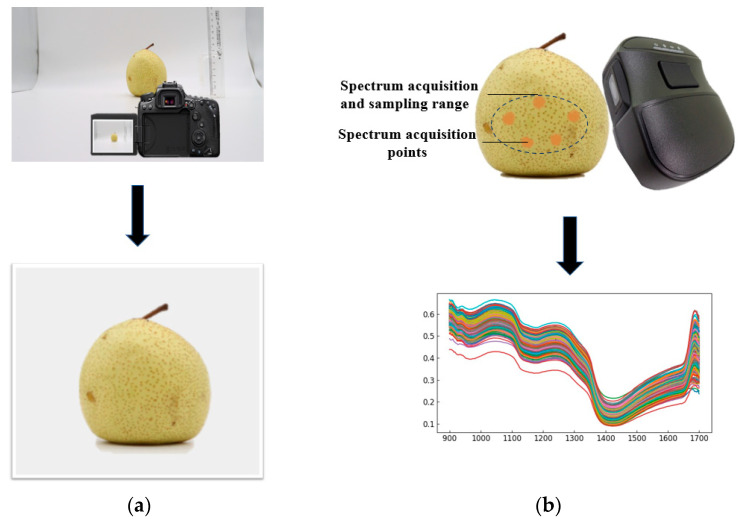
Sample collection process: (**a**) spectral data acquisition of samples; (**b**) image data acquisition of samples.

**Figure 4 foods-12-01178-f004:**
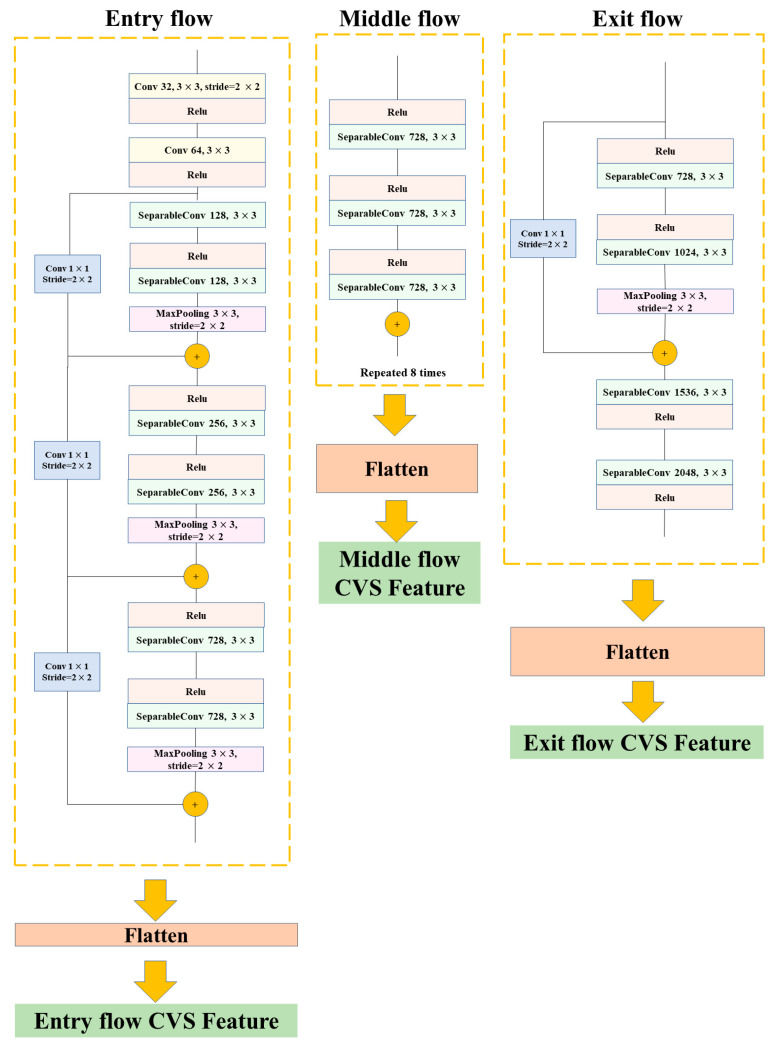
The architecture of the different layers of Xception.

**Figure 5 foods-12-01178-f005:**
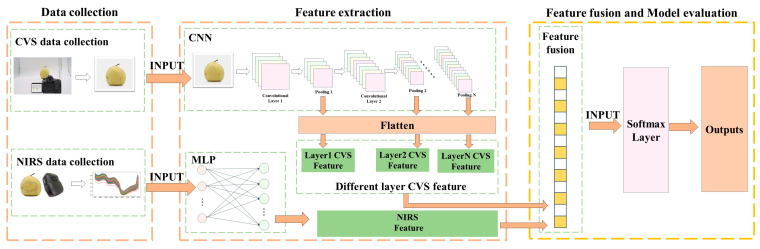
Description of spectral and image feature depth fusion methods.

**Figure 6 foods-12-01178-f006:**
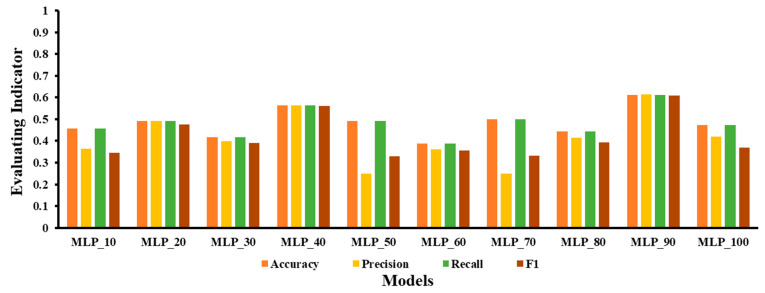
Plot of the validation set results when using MLP with different numbers of hidden layer nodes to build a classification model.

**Figure 7 foods-12-01178-f007:**
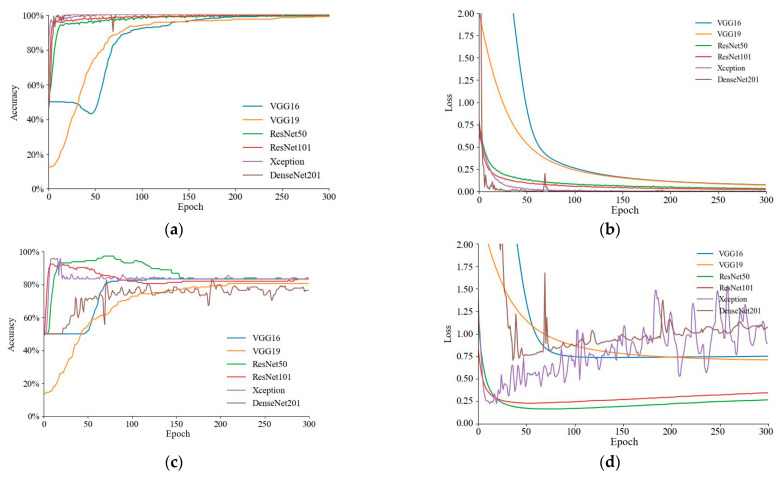
Training curves for six image feature classification models: (**a**) training accuracy curve; (**b**) training loss curve; (**c**) validation accuracy curve; and (**d**) validation loss curve.

**Figure 8 foods-12-01178-f008:**
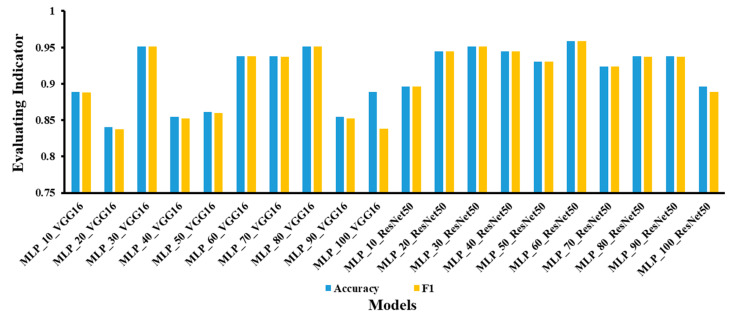
MLP model built with ten sets of hidden layer node numbers with VGG16 and MLP_ResNet50 fusion modeling of accuracy and F1.

**Figure 9 foods-12-01178-f009:**
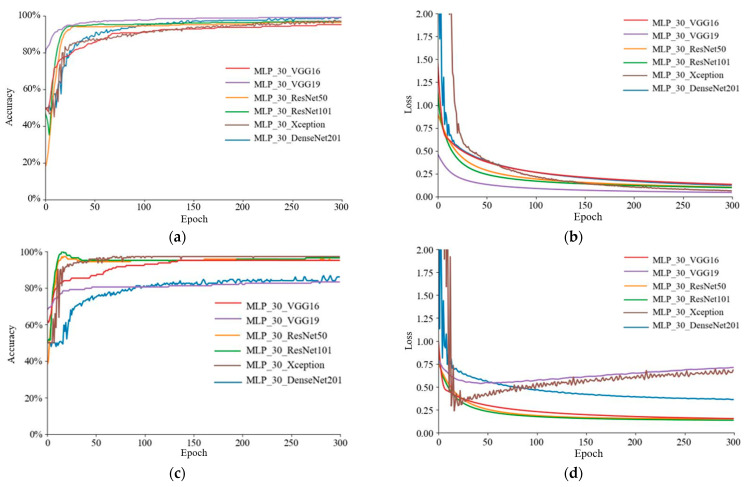
Training curves for six fusion models: (**a**) training accuracy curve; (**b**) training loss curve; (**c**) validation accuracy curve, and (**d**) validation loss curve.

**Figure 10 foods-12-01178-f010:**
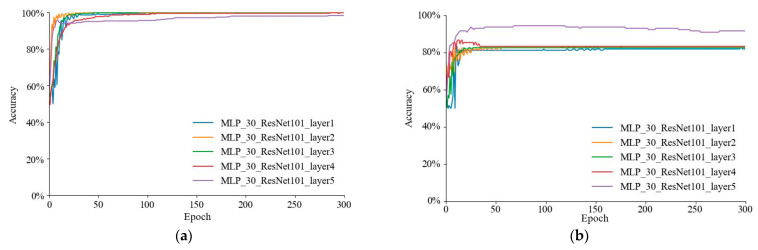
Results of different layer fusion modeling of convolutional neural networks: (**a**) training accuracy curves for five different layers of ResNet101; (**b**) validation accuracy curves for five different layers of ResNet101; (**c**) training accuracy curves for three different layer fusion modeling of Xception; and (**d**) validation accuracy curves for modeling of three different layer fusions of Xception.

**Figure 11 foods-12-01178-f011:**
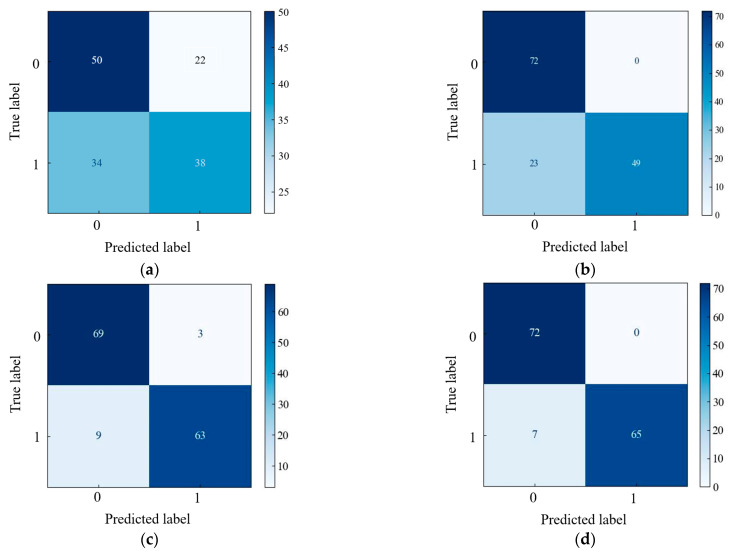
Accuracy confusion matrix for different models in the prediction set of the four optimal models: (**a**) MLP_90; (**b**) Xception; (**c**) MLP_ResNet101_layer5; and (**d**) MLP_30_Xception_Exitflow.

**Table 1 foods-12-01178-t001:** Structure and output size of the different layers of ResNet101.

No. Layer	Layer Name	Net	Output
1	Conv1	7 × 7, 64, stride2	112 × 112
2	Conv2_x	3 × 3 max pool, stride2 $\left[\begin{array}{c} 1\times 1,64\\ 3\times 3,64\\ 1\times 1,64 \end{array}\right]\times 3$	56 × 56
3	Conv3_x	$\left[\begin{array}{c} 1\times 1,128\\ 3\times 3,128\\ 1\times 1,512 \end{array}\right]\times 4$	28 × 28
4	Conv4_x	$\left[\begin{array}{c} 1\times 1,256\\ 3\times 3,128\\ 1\times 1,1024 \end{array}\right]\times 23$	14 × 14
5	Conv5_x	$\left[\begin{array}{c} 1\times 1,512\\ 3\times 3,512\\ 1\times 1,2048 \end{array}\right]\times 3$	7 × 7
		Average pool, 1000-d fc, SoftMax	1 × 1

**Table 2 foods-12-01178-t002:** Details of the parameterization used to adjust each final classification model.

Models	Parameters
PLS_DA	n_components = 8.
MLP	Neurons in hidden layers: 90; activation: Rectified Linear Unit (Relu); solver: Adam
SVM	C = 601, gamma = 0.15; kernel = “poly”
Random Forest	Limit the maximal tree depth:20; Number of trees: 15; Do not split subsets smaller than: 3.
AdaBoost	Number of estimators: 50; learning rate: 1.0; classification algorithm: SAMME.R; Regression loss function: Square.
XGBoost	Number of estimators: 50; learning rate: 1.0; classification algorithm: SAMME.R; Regression loss function: Square.

**Table 3 foods-12-01178-t003:** The validation set results for spectral feature classification models.

Models	Accuracy	Precision	Recall	F1
PLS_DA	0.597	0.601	0.597	0.593
MLP	0.611	0.614	0.611	0.608
SVM	0.507	0.510	0.507	0.468
Random Forest	0.403	0.393	0.403	0.389
AdaBoost	0.451	0.428	0.451	0.403
XGBoost	0.340	0.318	0.340	0.320

**Table 4 foods-12-01178-t004:** The validation set results for six image feature classification models.

Models	Accuracy	Precision	Recall	F1
VGG16	0.833	0.875	0.833	0.829
VGG19	0.806	0.831	0.806	0.802
ResNet50	0.833	0.875	0.833	0.829
ResNet101	0.833	0.875	0.833	0.829
Xception	0.840	0.879	0.840	0.836
DenseNet201	0.764	0.769	0.764	0.763

**Table 5 foods-12-01178-t005:** The validation set results for six fusion models.

Models	Accuracy	Precision	Recall	F1
MLP_30_VGG16	0.951	0.956	0.951	0.951
MLP_30_VGG19	0.833	0.875	0.833	0.829
MLP_30_ResNet50	0.951	0.952	0.951	0.951
MLP_30_ResNet101	0.965	0.966	0.965	0.965
MLP_30_Xception	0.972	0.974	0.972	0.972
MLP_30_DenseNet201	0.861	0.862	0.861	0.861

**Table 6 foods-12-01178-t006:** The validation set results for modeling the fusion of different layers of convolutional networks with spectral features.

Models	Accuracy	Precision	Recall	F1
MLP_30_ResNet101_layer1	0.819	0.852	0.819	0.815
MLP_30_ResNet101_layer2	0.833	0.875	0.833	0.829
MLP_30_ResNet101_layer3	0.826	0.864	0.826	0.822
MLP_30_ResNet101_layer4	0.833	0.875	0.833	0.829
MLP_30_ResNet101_layer5	0.917	0.920	0.951	0.917
MLP_30_Xception_Entryflow	0.833	0.875	0.833	0.829
MLP_30_Xception_Middleflow	0.792	0.853	0.792	0.782
MLP_30_Xception_Exitflow	0.951	0.956	0.951	0.951

**Table 7 foods-12-01178-t007:** Results of the optimal model built with different features on the validation set.

Characteristic Category	Models	Accuracy	Precision	Recall	F1
NIRS	MLP_90	0.611	0.614	0.611	0.608
CVS	Xception	0.840	0.879	0.840	0.836
Fusion Feature	MLP_30_ResNet101_layer5	0.917	0.920	0.951	0.917
Fusion Feature	MLP_30_Xception_Exitflow	0.951	0.956	0.951	0.951

## Data Availability

The datasets generated for this study are available on request to the corresponding author.

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
