# Peer review of "Identifying the “Dangshan” Physiological Disease of Pear Woolliness Response via Feature-Level Fusion of Near-Infrared Spectroscopy and Visual RGB Image"

_foods, 2023, doi:10.3390/foods12061178_

Round 1
Reviewer 1 Report
Dear authors,
I think the paper you wrote is rather interesting, with very extensive and robust method development, with persuasive conclusions. However, I would advise some improvements in the method section, when clarification is needed in few places, and I would also advice the language check. Please find the annotated pdf with my comments/suggestions where minor revisions are needed.
Best regards

Reviewer 2 Report
The manuscript is well written and reports a novel study to identify the Dangshan diease in pears by using NIR-RGB imaging with mid-high level data fusion. Despite the study was well performed, there are some revisions required:
1. Please provide the full expansion for “CSV” in the abstract.
2. Please report the results with only 3 decimal places (accuracy, precision, recall, F1) throughout the manuscript.
3. Introduction: please replace the word “work” by “study”.
4. Have the authors performed simpler classifiers such as partial least squares discriminant analysis (PLS-DA)?
5. Page 3, l. 138. Please specify the exact number of samples used in the study.
6. Section 2.2, l. 153. Please add the spectrometer brand and manufacturer.
7. Often the NIR spectral data are pre-processed to correct for variations caused by light scattering or random noise? Did the authors perform such pre-processing? This could improve the results for the NIR data only. Please clarify in the text.
8. I think figs. 2 and 3 could be combined into a single figure. Also, figs. 4 and 5 could be excluded.
9. Table 3. Please clarify in the table legend for what data (NIR or CSV) these models were calculated.
10. Please highlight the best results obtained in the conclusion.
